# SELF-SUPERVISED VARIATIONAL AUTO-ENCODERS

## ABSTRACT

Density estimation, compression, and data generation are crucial tasks in artificial intelligence. Variational Auto-Encoders (VAEs) constitute a single framework to achieve these goals. Here, we present a novel class of generative models, called *self-supervised Variational Auto-Encoder* (selfVAE), that utilizes deterministic and discrete transformations of data. This class of models allows performing both conditional and unconditional sampling while simplifying the objective function. First, we use a single self-supervised transformation as a latent variable, where a transformation is either downscaling or edge detection. Next, we consider a hierarchical architecture, i.e., multiple transformations, and we show its benefits compared to the VAE. The flexibility of selfVAE in data reconstruction finds a particularly interesting use case in data compression tasks, where we can trade-off memory for better data quality, and vice-versa. We present the performance of our approach on three benchmark image data (Cifar10, Imagenette64, and CelebA).

## 1 INTRODUCTION

The framework of variational autoencoders (VAEs) provides a principled approach for learning latent-variable models. As it utilizes a meaningful low-dimensional latent space with density estimation capabilities, it forms an attractive solution for generative modelling tasks. However, its performance in terms of the test log-likelihood and quality of generated samples is often disappointing, thus, many modifications were proposed. In general, one can obtain a tighter lower bound, and, thus, a more powerful and flexible model, by advancing over the following three components: the *encoder* (Rezende et al., 2014; van den Berg et al., 2018; Hoogeboom et al., 2020; Maaløe et al., 2016), the *prior* (or *marginal* over latents) (Chen et al., 2016; Habibian et al., 2019; Lavda et al., 2020; Lin & Clark, 2020; Tomczak & Welling, 2017) and the *decoder* (Gulrajani et al., 2016). Recent studies have shown that by employing deep hierarchical architectures and by carefully designing building blocks of the neural networks, VAEs can successfully model high-dimensional data and reach state-of-the-art test likelihoods (Zhao et al., 2017; Maaløe et al., 2019; Vahdat & Kautz, 2020).

In this work, we present a novel class of VAEs, called *self-supervised Variational Auto-Encoders*, where we introduce additional variables to VAEs that result from discrete and deterministic transformations of observed images. Since the transformations are deterministic, and they provide a specific aspect of images (e.g., contextual information through detecting edges or downscaling), we refer to them as *self-supervised representations*. The introduction of the discrete and deterministic variables allows to train deep hierarchical models efficiently by decomposing the task of learning a highly complex distribution into training smaller and conditional distributions. In this way, the model allows to integrate the prior knowledge about the data, but still enables to synthesize unconditional samples. Furthermore, the discrete and deterministic variables could be used to conditionally reconstruct data, which could be of great use in data compression and super-resolution tasks.

We make the following contributions: i) We propose an extension of the VAE framework by incorporating self-supervised representations of the data. ii) We analyze the impact of modelling natural images with different data transformations as self-supervised representations. iii) This new type of generative model (*self-supervised Variational Auto-Encoders*), which is able to perform both conditional and unconditional sampling, demonstrate improved quantitative performance in terms of density estimation and generative capabilities on image benchmarks.

## 2 BACKGROUND

### 2.1 VARIATIONAL AUTO-ENCODERS

Let $\mathbf{x} \in \mathcal{X}^D$ be a vector of observable variables, where $\mathcal{X} \subseteq \mathbb{R}$ or $\mathcal{X} \subseteq \mathbb{Z}$, and $\mathbf{z} \in \mathbb{R}^M$ denote a vector of latent variables. Since calculating $p_\vartheta(\mathbf{x}) = \int p_\vartheta(\mathbf{x}, \mathbf{z}) \mathrm{d}\mathbf{z}$ is computationally intractable for non-linear stochastic dependencies, a variational family of distributions could be used for approximate inference. Then, the following objective function could be derived, namely, the *evidence lower bound* (ELBO) (Jordan et al., 1999):

$$\ln p_\vartheta(\mathbf{x}) \geq \mathbb{E}_{q_\phi(\mathbf{z}|\mathbf{x})} \left[ \ln p_\theta(\mathbf{x}|\mathbf{z}) + \ln p_\lambda(\mathbf{z}) - \ln q_\phi(\mathbf{z}|\mathbf{x}) \right], \tag{1}$$

where $q_\phi(\mathbf{z}|\mathbf{x})$ is the variational posterior (or the *encoder*), $p_\theta(\mathbf{x}|\mathbf{z})$ is the conditional likelihood function (or the *decoder*) and $p_\lambda(\mathbf{z})$ is the *prior* (or *marginal*), $\phi$, $\theta$ and $\lambda$ denote parameters.

The expectation is approximated by Monte Carlo sampling while exploiting the *reparameterization trick* in order to obtain unbiased gradient estimators. The models are parameterized by neural networks. This generative framework is known as *Variational Auto-Encoder* (VAE) (Kingma & Welling, 2013; Rezende et al., 2014).

### 2.2 VAEs WITH BIJECTIVE PRIORS

Even though the lower-bound suggests that the prior plays a crucial role in improving the variational bounds, usually a fixed distribution is used, e.g., a standard multivariate Gaussian. While being relatively simple and computationally cheap, the fixed prior is known to result in over-regularized models that tend to ignore most of the latent dimensions (Burda et al., 2015; Hoffman & Johnson, 2016; Tomczak & Welling, 2017). Moreover, even with powerful encoders, VAEs may still fail to match the variational posterior to a unit Gaussian prior (Rosca et al., 2018).

However, it is possible to obtain a rich, multi-modal prior distribution $p(\mathbf{z})$ by using a *bijective* (or *flow-based*) model (Dinh et al., 2016). Formally, given a latent code $\mathbf{z}$, a base distribution $p_V(\mathbf{v})$ over latent variables $\mathbf{v} \in \mathbb{R}^M$, and $f : \mathbb{R}^M \to \mathbb{R}^M$ consisting of a sequence of $L$ diffeomorphic transformations[1], where $f_i(\mathbf{v}_{i-1}) = \mathbf{v}_i$, $\mathbf{v}_0 = \mathbf{v}$ and $\mathbf{v}_L = \mathbf{z}$, the *change of variable* can be used sequentially to express the distribution of $\mathbf{z}$ as a function of $\mathbf{v}$ as follows:

$$\log p(\mathbf{z}) = \log p_V(\mathbf{v}) - \sum_{i=1}^{L} \log \left| \frac{\partial f_i(\mathbf{v}_{i-1})}{\partial \mathbf{v}_{i-1}} \right|, \tag{2}$$

where $\left| \frac{\partial f_i(\mathbf{v}_{i-1})}{\partial \mathbf{v}_{i-1}} \right|$ is the Jacobian-determinant of the $i^{th}$ transformation.

Thus, using the bijective prior yields the following lower-bound:

$$\ln p(\mathbf{x}) \geq \mathbb{E}_{q_\phi(\mathbf{z}|\mathbf{x})} \left[ \log p_\theta(\mathbf{x}|\mathbf{z}) - \log q_\phi(\mathbf{z}|\mathbf{x}) + \log p_V(\mathbf{v}_0) + \sum_{i=1}^{L} \log \left| \frac{\partial f_i^{-1}(\mathbf{v}_i)}{\partial \mathbf{v}_i} \right| \right]. \tag{3}$$

In this work, we utilize RealNVP (Dinh et al., 2016) as the prior, however, any other flow-based model could be used (Kingma & Dhariwal, 2018; Hoogeboom et al., 2020). For the experiments and ablation study that shows the impact of the bijective prior on VAEs, we refer to the appendix A.1.

## 3 METHOD

### 3.1 MOTIVATION

The idea of self-supervised learning is about utilizing original unlabeled data to create additional context information. It could be achieved in multiple manners, e.g., by adding noise to data (Vincent et al., 2008) or masking data during training (Zhang et al., 2017). Self-supervised learning could also be seen as turning an unsupervised model into a supervised by, e.g., treating predicting next pixels as a classification task (Hénaff et al., 2019; Oord et al., 2018). These are only a few examples of a quickly growing research line (Liu et al., 2020).

---

[1]That is, invertible and differentiable transformations.

Here, we propose to use non-trainable transformations to obtain information about image data. Our main hypothesis is that since working with highly-quality images is challenging, we could alleviate this problem by additionally considering partial information about them. Fitting a model to images of lower quality, and then enhancing them to match the target distribution seems to be overall an easier task (Chang et al., 2004; Gatopoulos et al., 2020). By incorporating compressed transformations (i.e., the self-supervised representations) that still contain global information, with the premise that it would be easier to approximate, the process of modelling a high-dimensional complex density breaks down into simpler tasks. In this way, the expressivity of the model will grow and gradually result into richer, better generations. A positive effect of the proposed framework is that the model allows us to integrate prior knowledge through the image transformations, without losing its uncon-

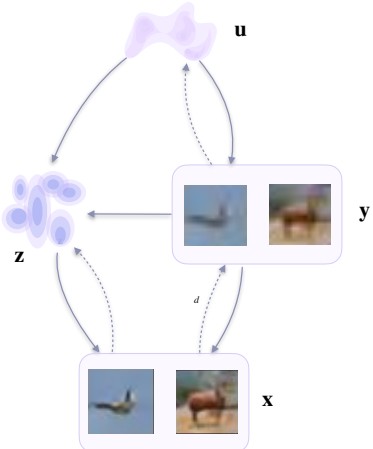

Figure 1: The proposed approach.

ditional generative functionality. Overall, we end up with a two-level VAE with three latent variables, where one is a data transformation that can be obtained in a self-supervised fashion. In Figure 1 a schematic representation of the proposed approach with downscaling is presented.

A number of exemplary image transformations are presented in Figure 2. We notice that with these transformations, even though they discard a lot of information, the global structure is preserved. As a result, in practice the model should have the ability to extract a general concept of the data, and add local information afterwards. In this work, we focus on downscaling (Figure 2.b, c & d) and edge detection or *sketching* (Fig. 2.i).

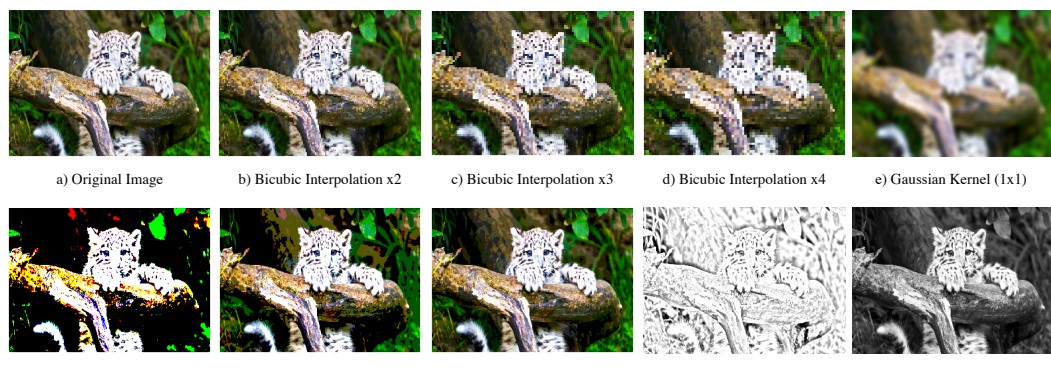

Figure 2: Image Transformations. All of these transformations still preserve the global structure of the samples but they disregard the high resolution details in different ways.

### 3.2 MODEL FORMULATION

In our model, we consider representations that result from *deterministic* and *discrete* transformations of an image. Formally, we introduce a transformation $d : \mathcal{X}^D \to \mathcal{X}^C$ that takes $\mathbf{x}$ and returns an image representation $\mathbf{y}$, e.g., a downscaled image. Since we lose information about the original image, $\mathbf{z}$ could be seen as a variable that compensates lost details in $\mathbf{x}$. Further we propose to introduce an additional latent variable, $\mathbf{u} \in \mathbb{R}^N$ to model $\mathbf{y}$ and $\mathbf{z}$. We can define the joint distribution of $\mathbf{x}$ and $\mathbf{y}$ as follows: $p(\mathbf{x}, \mathbf{y}) = p(\mathbf{y}|\mathbf{x})p(\mathbf{x})$, where $p(\mathbf{y}|\mathbf{x}) = \delta(\mathbf{y} - d(\mathbf{x}))$ due to the deterministic transformation $d(\cdot)$, where $\delta(\cdot)$ is the Kronecker delta. Thus, the empirical distribution is $\delta(\mathbf{y} - d(\mathbf{x}))p_{data}(\mathbf{x})$. However, since we are interested in decomposing the problem of modeling a complex distribution $p(\mathbf{x})$, we propose to model $p(\mathbf{x}|\mathbf{y})p(\mathbf{y})$ instead, and utilize the variational inference of the form $Q(\mathbf{u}, \mathbf{z}|\mathbf{x}, \mathbf{y}) = q(\mathbf{u}|\mathbf{y})q(\mathbf{z}|\mathbf{x})$ that yields:

$$\ln p(\mathbf{x}, \mathbf{y}) \geq \mathbb{E}_Q\big[\ln p_\theta(\mathbf{x}|\mathbf{y}, \mathbf{z}) + \ln p(\mathbf{z}|\mathbf{u}, \mathbf{y}) + \ln p(\mathbf{y}|\mathbf{u}) + \ln p(\mathbf{u}) - \ln q(\mathbf{z}|\mathbf{x}) - \ln q(\mathbf{u}|\mathbf{y})\big]. \quad (4)$$

Intuitively, the premise for selfVAE is that the latents $\mathbf{u}$ will capture the global structure of the input data and the latents $\mathbf{z}$ will encode the missing information between $\mathbf{y}$ and $\mathbf{x}$, guiding the model to discover the distribution of the target observations. In order to highlight the self-supervised part in our model, we refer to it as the *self-supervised Variational Auto-Encoder* (or selfVAE for short).

Further, we propose to choose the following distributions:

$$p(\mathbf{v}) = \mathcal{N}\left(\mathbf{v}|\mathbf{0}, \mathbf{1}\right) \qquad\qquad q_{\phi_1}\left(\mathbf{u}|\mathbf{y}\right) = \mathcal{N}\left(\mathbf{u}|\boldsymbol{\mu}_{\phi_1}(\mathbf{y}), \mathrm{diag}\left(\boldsymbol{\sigma}_{\phi_1}(\mathbf{y})\right)\right)$$

$$p_\lambda\left(\mathbf{u}\right) = p(\mathbf{v}) \prod_{i=1}^{F} \left| \det \frac{\partial f_i(\mathbf{v}_{i-1})}{\partial \mathbf{v}_{i-1}} \right|^{-1} \qquad q_{\phi_2}\left(\mathbf{z}|\mathbf{x}\right) = \mathcal{N}\left(\mathbf{z}|\boldsymbol{\mu}_{\phi_2}(\mathbf{x}), \mathrm{diag}\left(\boldsymbol{\sigma}_{\phi_2}(\mathbf{x})\right)\right).$$

$$p_{\theta_1}\left(\mathbf{y}|\mathbf{u}\right) = \sum_{i=1}^{I} \pi_i^{(\mathbf{u})} \mathrm{Dlogistic}\left(\mu_i^{(\mathbf{u})}, s_i^{(\mathbf{u})}\right)$$

$$p_{\theta_2}\left(\mathbf{z}|\mathbf{y}, \mathbf{u}\right) = \mathcal{N}\left(\mathbf{z}|\boldsymbol{\mu}_{\theta_2}(\mathbf{y}, \mathbf{u}), \mathrm{diag}\left(\boldsymbol{\sigma}_{\theta_2}(\mathbf{y}, \mathbf{u})\right)\right)$$

$$p_{\theta_3}\left(\mathbf{x}|\mathbf{z}, \mathbf{y}\right) = \sum_{i=1}^{I} \pi_i^{(\mathbf{z}, \mathbf{y})} \mathrm{Dlogistic}\left(\mu_i^{(\mathbf{z}, \mathbf{y})}, s_i^{(\mathbf{z}, \mathbf{y})}\right)$$

where Dlogistic is defined as the discretized logistic distribution (Salimans et al., 2017), and we utilize a flow-based model for $p_\lambda\left(\mathbf{u}\right)$. Notice that we use the discretized logistic distribution, because images are represented by values between 0 and 255. For integer-valued random variables, other distributions like Gaussian are inappropriate.

### 3.3 GENERATION AND RECONSTRUCTION IN SELFVAE

As generative models, VAEs can be used to synthesize novel content through the following process: $\mathbf{z} \sim p(\mathbf{z}) \rightarrow \mathbf{x} \sim p(\mathbf{x}|\mathbf{z})$, but also to reconstruct a data sample $\mathbf{x}^*$ by using the following scheme: $\mathbf{z} \sim q(\mathbf{z}|\mathbf{x}^*) \rightarrow \mathbf{x} \sim p(\mathbf{x}|\mathbf{z})$.

Interestingly, our approach allows to utilize more operations regarding data generation and reconstruction. First, analogously to VAEs, the selfVAE allows to generate data by applying the following hierarchical sampling process (*generation*):

$$\mathbf{u} \sim p(\mathbf{u}) \rightarrow \mathbf{y} \sim p(\mathbf{y}|\mathbf{u}) \rightarrow \mathbf{z} \sim p(\mathbf{z}|\mathbf{u}, \mathbf{y}) \rightarrow \mathbf{x} \sim p(\mathbf{x}|\mathbf{y}, \mathbf{z}).$$

However, we can use the ground-truth $\mathbf{y}$ (i.e, $\mathbf{y}^* = d(\mathbf{x}^*)$), and sample or infer $\mathbf{z}$. Then, the generative process for the former (*conditional generation*) is:

$$\mathbf{z} \sim q(\mathbf{z}|\mathbf{x}^*) \rightarrow \mathbf{x} \sim p(\mathbf{x}|\mathbf{y}^*, \mathbf{z}),$$

and for the latter (*conditional reconstruction*):

$$\mathbf{u} \sim q(\mathbf{u}|\mathbf{y}^*) \rightarrow \mathbf{z} \sim p(\mathbf{z}|\mathbf{u}, \mathbf{y}^*), \rightarrow \mathbf{x} \sim p(\mathbf{x}|\mathbf{y}^*, \mathbf{z}).$$

If $\mathbf{y}$ is a downscaling transformation of the input image, selfVAE can be used in a manner similar to the super-resolution (Gatopoulos et al., 2020). Alternatively, we can sample (or generate) $\mathbf{y}$ instead, and choose to sample or infer $\mathbf{z}$. In this way, we can reconstruct an image in two ways, namely, *reconstruction 1*:

$$\mathbf{y}^* = d(\mathbf{x}^*) \rightarrow \mathbf{u} \sim q(\mathbf{u}|\mathbf{y}^*) \rightarrow \mathbf{y} \sim p(\mathbf{y}|\mathbf{u}) \rightarrow \mathbf{z} \sim p(\mathbf{z}|\mathbf{u}, \mathbf{y}) \rightarrow \mathbf{x} \sim p(\mathbf{x}|\mathbf{z}, \mathbf{y}),$$

and *reconstruction 2*:

$$\left(\mathbf{y}^* = d(\mathbf{x}^*) \rightarrow \mathbf{u} \sim q(\mathbf{u}|\mathbf{y}^*) \rightarrow \mathbf{y} \sim p(\mathbf{y}|\mathbf{u})\right), \text{ then } \mathbf{z} \sim q(\mathbf{z}|\mathbf{x}^*) \rightarrow \mathbf{x} \sim p(\mathbf{x}|\mathbf{y}, \mathbf{z}).$$

The presented versions of generating and reconstructing images could be useful in the compression task. As we will see in the experiments, each option creates a different ratio of the reconstruction quality against the memory that we need to allocate to send information. However, every inferred variable needs to be sent, thus, more sampling corresponds to lower memory requirements.

### 3.4 Hierarchical self-supervised VAE

The proposed approach can be further extended and generalized by introducing multiple transformations, in the way that it is illustrated in Figure 3. By incorporating a single (or multiple) self-supervised representation(s) of the data, the process of modelling a high-dimensional complex density breaks down into $K$ simpler modeling tasks. Thus, we obtain a $K$-level VAE architecture, where the overall expressivity of the model grows even further and gradually results into generations of higher quality. Some transformations cannot be applied multiple times (e.g., edge detection), however, others could be used sequentially, e.g., downscaling.

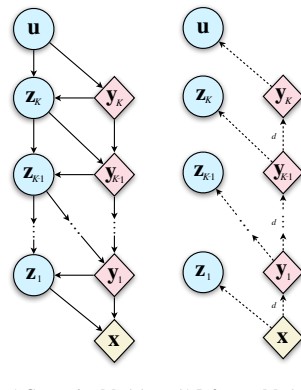

a) Generative Model    b) Inference Model

Figure 3: Hierarchical selfVAE.

We take $K$ self-supervised data transformations $d_k(\cdot)$ that give $K$ representations denoted by $\mathbf{y}_{1:K} = [\mathbf{y}_1, \ldots, \mathbf{y}_K]$, and the following variational distributions:

$$Q(\mathbf{u}, \mathbf{z}|\mathbf{x}, \mathbf{y}_{1:K}) = q(\mathbf{u}|\mathbf{y}_K)q(\mathbf{z}_1|\mathbf{x}) \prod_{k=1}^{K-1} q(\mathbf{z}_{k+1}|\mathbf{y}_k), \tag{5}$$

that yields the following objective:

$$\ln p(\mathbf{x}, \mathbf{y}_{1:K}) \geq \mathbb{E}_Q \Big[ \ln p_\theta(\mathbf{x}|\mathbf{y}_1, \mathbf{z}_1) + \sum_{k=1}^{K-1} \big( \ln p(\mathbf{z}_k|\mathbf{y}_k, \mathbf{z}_{k+1}) + \ln p(\mathbf{y}_k|\mathbf{y}_{k+1}, \mathbf{z}_{k+1}) \big) +$$

$$+ \ln p(\mathbf{z}_K|\mathbf{u}, \mathbf{y}_K) + \ln p(\mathbf{y}_K|\mathbf{u}) + \ln p(\mathbf{u}) - \ln q(\mathbf{u}|\mathbf{y}_K) - \ln q(\mathbf{z}_1|\mathbf{x}) - \sum_{k=1}^{K-1} \ln q(\mathbf{z}_{k+1}|\mathbf{y}_k) \Big]. \tag{6}$$

## 4 Experiments

### 4.1 Experimental Setup

**Datasets**    We evaluate the proposed model on CIFAR-10, Imagenette64 and CelebA:

*CIFAR-10*    The CIFAR-10 dataset is a well-known image benchmark data containing 60.000 training examples and 10.000 validation examples. From the training data, we put aside $15\%$ randomly selected images as the test set. We augment the training data by using random horizontal flips and random affine transformations and normalize the data uniformly in the range (0, 1).

*Imagenette64*    Imagenette64[2] is a subset of 10 classes from the downscaled Imagenet dataset. We downscaled the dataset to 64px $\times$ 64px images. Similarly to CIFAR-10, we put aside $15\%$ randomly selected training images as the test set. We used the same data augmentation as in CIFAR-10

*CelebA*    The Large-scale CelebFaces Attributes (CelebA) Dataset consists of 202.599 images of celebrities. We cropped original images on the 40 vertical and 15 horizontal component of the top left corner of the crop box, which height and width were cropped to 148. Besides the uniform normalization of the image, no other augmentation was applied.

**Architectures**    Encoders and decoders consist of building blocks composed of DenseNets (Huang et al., 2016), channel-wise attention (Zhang et al., 2018), and ELUs (Clevert et al., 2015) as activation functions. The dimensionality of all the latent variables were kept at $8 \times 8 \times 16 = 1024$ and all models were trained using AdaMax (Kingma & Ba, 2014) with data-dependent initialization (Salimans & Kingma, 2016). Regarding the selfVAEs, in CIFAR-10 we used an architecture with a single downscaled transformation (selfVAE-downscale), while on the remaining two datasets (CelebA

---

[2]https://github.com/fastai/imagenette

and Imagenette64) we used a hierarchical 3-leveled selfVAE with downscaling, and a selfVAE with sketching. All models were employed with the bijective prior (RealNVP) comparable in terms of the number of parameters (the range of the weights of all models was from 32M to 42M). For more details, please refer to the appendix section A.2.

**Evaluation**    We approximate the negative log-likelihood using 512 IW-samples (Burda et al., 2015) and express the scores in bits per dimension (*bpd*). Additionally, for CIFAR-10, we use the *Fréchet Inception Distance* (FID) (Heusel et al., 2017).

Table 1: Quantitative comparison on test sets from CIFAR-10, CelebA, and Imagenette64. *Measured on training set.

| Dataset | Model | $bpd \downarrow$ | FID $\downarrow$ |
|---|---|---|---|
| CIFAR-10 | PixelCNN (van den Oord et al., 2016) | 3.14 | 65.93 |
| | GLOW (Kingma & Dhariwal, 2018) | 3.35 | 65.93 |
| | ResidualFlow (Chen et al., 2019) | 3.28 | 46.37 |
| | BIVA (Maaløe et al., 2019) | 3.08 | - |
| | NVAE (Vahdat & Kautz, 2020) | **2.91** | - |
| | DDPM (Ho et al., 2020) | 3.75 | **5.24** (**3.17**\*) |
| | VAE (ours) | 3.51 | 41.36 (37.25\*) |
| | selfVAE-downscale | 3.65 | 34.71 (29.95\*) |
| CelebA | RealNVP (Dinh et al., 2016) | 3.02 | - |
| | VAE (ours) | 3.12 | - |
| | selfVAE-sketch | 3.24 | - |
| | selfVAE-downscale-3lvl | **2.88** | - |
| Imagenette64 | VAE (ours) | 3.85 | - |
| | selfVAE-downscale-3lvl | **3.70** | - |

## 4.2    QUANTITATIVE RESULTS

We present the results of the experiments on the benchmark datasets in Table 1. First, we notice that on CIFAR-10 our implementation of the VAE is still lacking behind other generative models in terms of *bpd*, however, it is better or comparable in terms of FID. The selfVAE-downscale achieves worse *bpd* than the VAE. A possible explanation may lie in the small image size ($32 \times 32$), as the benefits of breaking down the learning process in two or more steps are not obvious given the small target dimensional space. Nevertheless, the selfVAE-downscale achieves significantly better FID scores than the other generative models. This result could follow from the fact that downscaling allows to maintain context information about the original image and, as a result, a general coherence is of higher quality.

Interestingly, on the two other datasets, a three-level selfVAE-downscale achieves significantly better *bpd* scores than the VAE with the bijective prior. This indicates the benefit of employing a multi-leveled self-supervised framework against the VAE in higher-dimensional data, where the plain model fails to scale efficiently. It seems that the hierarchical structure of self-supervised random variables allows to encode the missing information more efficiently in $\mathbf{z}_k$, in contrast to the vanilla VAE, where all information about images must be coded in $\mathbf{z}$. This result is promising and indicates that the proposed approach is of great potential for generative modelling.

## 4.3    QUALITATIVE RESULTS

We present generations on CIFAR-10 and Imagenette64 in Figure 4 and on CelebA in Figure 5, and reconstructions on CIFAR-10 and CelebA in Figure 6.

selfVAE - downscale        VAE

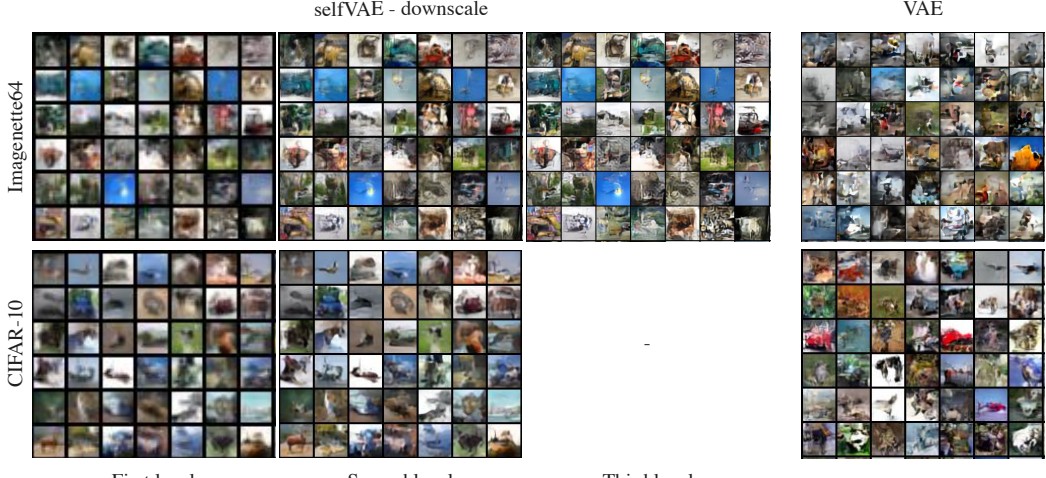

First level        Second level        Third level

Figure 4: Uncoditional generations on Imagenette64 and CIFAR-10.

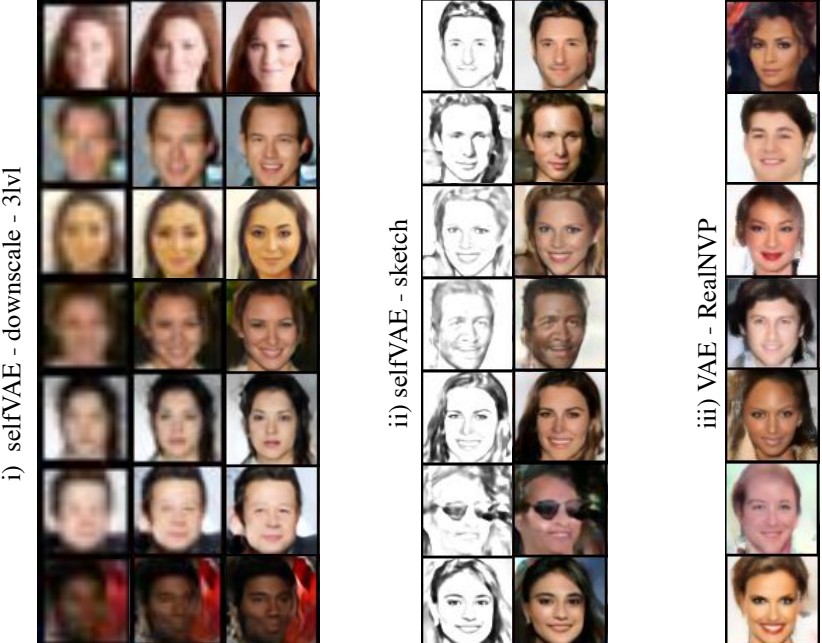

Figure 5: Unconditional CelebA generations from (i) the three-level self-supervised VAE with downscaling, (ii) the self-supervised VAE employed with edge detection (sketches), (iii) the VAE with RealNVP prior.

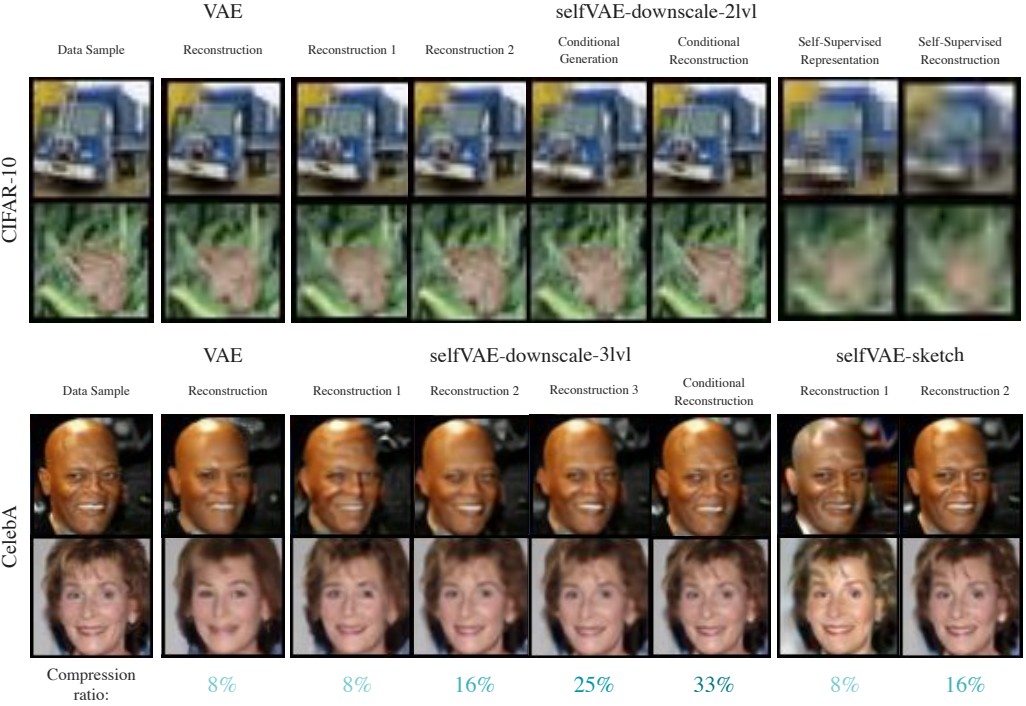

Figure 6: Comparison on image reconstructions with different amount of sent information.

We first notice that the generations from selfVAE seem to be more coherent, in contrast with these from VAE that produces overall more contextless and distorted generations. This result seems to be in line with the FID scores. Especially for CelebA, we observe impressive synthesis quality, great sampling diversity and coherent generations (Figure 5). On the Imagenette64 dataset, we can also observe crisper generations for our method compared to the VAE (Figure 4).

Furthermore, the hierarchical selfVAE seems to be of a great potential for compression purposes. In contrast to the VAE, which is restricted to using a single way of reconstructing an image, the selfVAE allows four various options with different quality/memory ratios (Figure 6). In the selfVAE-sketch, we can retrieve the image with high accuracy by using only 16% of the original data, as it manages to encode all the texture of the image to **z** (Figure 11). This shows the advantage of choosing prior knowledge into the learning process. Lastly, the latents learn to add extra information, which defines the end result, and we can alter details of an image like facial expressions (Figure 12.ii).

## 5 CONCLUSION

In this paper, we showed that taking deterministic and discrete transformations results in coherent generations of high visual quality, and allows to integrate prior knowledge without loosing its unconditional generative functionality. The experimental results seem to confirm that hierarchical architectures perform better and allow to obtain both better *bpd* scores and better generations and reconstructions. In the experiments, we considered two classes of image transformations, namely, *downscaling* and edge detection (*sketching*). However, there is a vast of possible other transformations (see Figure 2), and we leave investigating them for future work. Moreover, we find the proposed approach interesting for the compression task. A similar approach with a multi-scale auto-encoder for image compression was proposed, e.g, by Mentzer et al. (2019) or Razavi et al. (2019). However, we still use a probabilistic framework and indicate that various non-trainable image transformations (not only multiple scales) could be of great potential.

## ACKNOWLEDGMENTS

Anonymized for the double-blind review.

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
