# OpenReview forum: "Self-Supervised Variational Auto-Encoders"
_ICLR.cc/2021/Conference — Reject_

### Official Review · AnonReviewer1 · 2020-10-26
**A interesting idea, but the results look not competitive**

**Rating:** 4
**Confidence:** 4

**Review:**

This paper focuses on the task of generating high-quality data with generative models. To be specific, the authors proposed a variant of variational autoencoder (VAE) model, named self-supervised VAE. The intuition behind this model is that by breaking down the complex generation task into simpler/smaller ones, complex models can be trained steadily with the guidance from the simpler-level task. To his end, a hierarchical generative model with multiple-level latent variables is proposed, in which lower-level latent variables are governed by lower-level data features. The lower-level feature is generally obtained by a determined and discrete transformation, like down scaling. In addition, to further the modeling capability, a flow-based prior is proposed to fit the data distribution. Experiments were conducted to evaluate the performance of the proposed generative model.


Strength:
1. The idea of guiding the complex image generation with easer tasks is interesting, and is maybe the right way to accomplish complex tasks.

2. the ELOB directed in Eq.2 is intuitive and insightful. It also provides me theoretical support for the fact that employing two-level modeling and downscale transformation to generate a more vivid image is reasonable.


Weakness:
1. From a technical perspective, the proposed method is just the combination of flow-based VAE and auxiliary VAE. By using 3 auxiliary variables, the authors infer one of them by a discrete and determined variational distribution q(y|x) to simplify the training objective, where the downscale image y plays an important role in this model. My question is why not regard y as observed data and then model the joint distribution p(x,y).

2. There are some mistakes in the derivation of Eq 2. In appendix A.4, during computing the entropy of q(w|x), the authors expresses it as E_{q(w|x)}[\log q(w|x)] = E_{q(z|y,x)}[\log q(z|y,x)] + ... . However, the first term in RHS is completely wrong. Actually, it should be E_{q(z|y,x)q(y|x)}[\log q(z|y,x)]. It seems that the authors use an equation in many places, that is E_{q(z|y,x)}[\log q(z|y,x)] = E_{q(z|y,x)q(u|y)q(y|x)}[\log q(z|y,x)]. But this equation is not ture, because the term q(z|y,x) insided the expectation is dependent on variable y. Besides, in the choice of distribution p(y|u) and p(x|z,y), they are set to be a mixture discrete logistic distribution. For each image x or y,  are their pixels assumed to be i.i.d ? If so, you miss \prod_{y_j \in y} outside the \sum_{i=1}^{I} in the distribution definition.

3. Bijective prior (RealNVP) is proposed in other works, and here simply employing it should not be regard as a contribution of this paper. Moreover, the authors only compare the effectiveness of different priors (i.e. Gaussian, mixture Gaussian, and RealNVP) on vanilla VAE and confirm the superiority of using an adaptive prior. However, I want to know what is the performance of self-supervised VAE if only using a standard Gaussian prior.

4. Section 3.3 is not presented well and the idea behind the sentences is hard to follow. What are the differences between these generation and reconstruction methods, and what application scenarios are corresponding to them? They are just simply listed, without providing any analysis of the logic behind them.

5. The experimental results cannot support the superiority of the proposed model in both of the quantity and quality comparisons. From the generated images, I cannot see too much difference between the SelfVAE and the vanilla VAE model, without to naming the more superior generative models, like GLOW, GANs etcs. Also, for the quantity comparison, the model is only compared with the outdated vanilla VAE in CelebA and Imagenet64, more recent generative models should be included here.

---

> ### Author Response · Authors · 2020-11-16
> **Response to AnonReviewer1**
>
> We would like to thank the reviewer for very interesting and insightful comments. They helped us to significantly improve the paper.
>
> **Point 1**
>
> We realized that after submitting the paper. It is true, it is not necessary to consider $y=d(x)$ as a part of a variational posterior. Instead, we can consider a joint distribution $p(x,y)$ that we can factorize as $p(y|x) p(x)$, where $p(y|x) = \delta(y - d(x))$. As a result, we can obtain both x’s and y’s as training data. However, while modeling, we are interested in the other factorization, namely, $p(x,y) = p(x|y) p(y)$, and then we can apply the variational inference with z’s and u’s as latent variables. We have rewritten the paper accordingly.
>
> **Point 2**
>
> Yes, we also realized this after submitting the paper. There were multiple errors. In the new version of the paper, we have corrected them.
>
> **Point 3**
>
> This is true, we have removed the awkward statement in the paragraph with contributions. At the very beginning of this project, we decided to first train a vanilla VAE that achieves bpd on CIFAR10 as close as possible to flow-based models (e.g., RealNVP). Therefore, we put a lot of effort into that. This included checking various architectures of encoders and decoders, as well as various priors. Once we determined the best performing model, we adapted it to the selfVAE while keeping a similar number of parameters for a fair comparison. As a result, we always trained the vanilla VAE first, and then, afterward, we trained the selfVAE.
> Our very initial experiments indicated that the difference between using a bijective prior vs. the standard Gaussian prior in selfVAE was similar to the same situation in a vanilla VAE. Therefore, we skipped carrying out similar experiments with selfVAE due to limited computational resources.
>
> **Point 4**
>
> In this section, we indeed list different manners of utilizing our approach for generating and reconstructing images. Since a vanilla VAE allows us to generate and reconstruct images in a single way, we wanted to highlight that incorporating self-supervised representations gives us more flexibility. In the new version of the paper, we added additional comments that these new ways of reconstructing open new perspectives for the compression task. Moreover, we have removed the corresponding figure, because we realized it could confuse a reader.
>
> **Point 5**
>
> Due to the double-blind policy, we cannot explain our situation in depth. However, we want to highlight that we had limited access to computational resources. We are aware that this is not a good excuse, therefore, we did our best to gather as much empirical evidence as possible. As a result, we have experimented on three datasets while two of them contain 64x64 images. We are almost certain that our approach would work even better on larger images. Nevertheless, at this point, we have no empirical evidence to support our belief.
> Unfortunately, we were unable to train larger models than 35-40M parameters. Currently, some papers report SOTA bpd for VAEs with over 100M parameters like NVAE or BIVA on CIFAR10. We include them in our comparison even though we cannot compare properly with them.
> We agree that providing only the VAE as a baseline for CelebA and Imagenette might be misleading. After an extensive literature search, we were able to find the bpd score for CelebA only in the RealNVP paper. Surprisingly, CelebA is widely used for the qualitative assessment, however, almost no-one provides the bpd. In the case of Imagenette, we are afraid that no-one else provides the bpd score (we were unable to find any paper).

---

### Official Review · AnonReviewer3 · 2020-10-26
**This paper describes a framework that combines Variational Autoencoders (VAE) with self-supervised transformations by adding latent variables such as downscaling and edge detection. The main idea is to match the latent distribution of the original and transformed (downscaled or edge detection) data. Experimental results are done on Cifar-10, ImageNet-64, and CelebA datasets.**

**Rating:** 4
**Confidence:** 5

**Review:**

###################################
Pros:

$\bullet$ VAEs can ignore some dimensions of the latent code. Enforcing the posterior distributions to consider desired factors of variations in the input can be fulfilled by either making it more structured (i.e., quantization as in VQ-VAE-2) or introducing additional constraints. This paper tackles this problem by applying the latter, two self-supervised tasks: edge maps and downscaled versions of inputs.

$\bullet$ The idea of adding self-supervised tasks to improve latent representation is very interesting. When learning a more structured latent representation, image superresolution, or sketch-to-image networks are also trained.

###################################
Cons:

$\bullet$ *Hierarchical self-supervised tasks* In section 3.4, multiple transformations explained. However, none of the experiments are conducted as a consecutive set of transformations. Does 3-level downscale mean a single downscaling three times or generating from $u$ using four different networks and match each of these levels with $z$? If yes, why does not the selfVAE-sketches model apply in a similar way hierarchically?

$\bullet$ *How to train/balance operations* In Figure 3, there are several modes of operations given. How did you balance these modes during training?

$\bullet$ *Performance of self-supervised tasks:* What is the effect of self-supervised tasks' performance on the quality of latent representations? Considering the literature in image superresolution and sketch-to-image, did you use a pretrained auxiliary generator?

$\bullet$ *What is RE and KL in Table?* Are they the summation of both reconstruction ($RE_x, RE_y$) and KL divergence ($KL_z, KL_u$) terms in the loss (Eq.2)?  The reason why previous methods' RE/KL values were omitted should be stated. Similarly, why were the FID scores on CelebA and ImageNet-64 not given?

Furthermore, the state-of-the-art FID scores on CIFAR-10 is better than the methods compared in Table 1. For instance, some examples of FID scores on CIFAR-10 are 18.9 in MoML [1], 29.3 in WP-GAN [2], 29.3 in spectrally normalized GAN [3], 26.4 in adversarial score matching [4], and so on.

[1] https://arxiv.org/pdf/1806.11006.pdf
[2] https://arxiv.org/pdf/1706.08500.pdf
[3] https://arxiv.org/pdf/1802.05957.pdf
[4] https://arxiv.org/pdf/2009.05475.pdf

As the results on CelebA and Imagenet-64 were not compared with previous literature, it is difficult to understand whether the contribution w.r.t. vanilla VAE is due to the self-supervised task or merely the use of an additional stochastic variable ($u$) and networks.

Minor issue: "Imagenette64" might cause confusion, I did not see this dataset name before. I suppose that it is "ImageNet resized to 64x64" as in PixelCNN paper. References for all datasets should be added.

###################################
Reasons for score:
Overall, I rate towards rejection. Even though the idea of bijective priors and doing this through self-supervised tasks is a novel approach, my major concern is that it is beyond the state-of-the-art in CIFAR-10, not compared to any other method on CelebA and ImageNet-64. Hopefully, the authors address my concerns above in the rebuttal period.

---

> ### Author Response · Authors · 2020-11-16
> **Response to AnonReviewer3**
>
> In general, we would like to thank you for your detailed comments. We realized that the paper contains errors and it is ambiguous in some parts. Therefore, we have rewritten it significantly and reformulated the equations.
>
> **Point Hierarchical self-supervised tasks**
>
> Both CelebA and Imagenette utilize a series of downscaling transformations (x: 64x64 -> y1: 32x32 -> y2: 16x16). Therefore, we used hierarchical selfVAE in these experiments.
> We did not use sketching in a hierarchical model, because sketching is equivalent to edge detection. Applying an edge detection transformation to an image consisting of detected edges does not provide any new information. However, it is possible to combine various transformations, e.g., downscaling and then sketching. Unfortunately, we did not have enough computational resources to carry out these experiments.
>
> **Point How to train/balance operations**
>
> We realized that this figure is rather confusing, and, therefore, we have removed it. The whole subsection on “Generation and Reconstruction in selfVAE” is not about training, it indicates that selfVAE allows us to perform reconstruction and generation in different ways. The training procedure is the same as in any other VAE, namely, we update weights of the encoder, the decoder, and the prior by optimizing the appropriate ELBO.
>
> **Point Performance of self-supervised tasks**
>
> We have tested all models (i.e., selfVAEs) on the density estimation task only. We do not have any experiments on tasks like super-resolution. It is an interesting research direction, and we are sure that our framework could provide a great platform for such tasks.
>
> **Point What is RE and KL in Table & FID scores**
>
> We realized that both RE and KL could be misleading. Therefore, we have removed them from the paper.
> We have used the WGAN since it is typically reported as a GAN-based baseline in other papers on the likelihood-based models. We did an additional search for a work that achieves state-of-the-art FID scores. As a result, we have replaced the WGAN with the following model:
> - Ho, J., Jain, A., & Abbeel, P. (2020). Denoising diffusion probabilistic models. Advances in Neural Information Processing Systems, 33.
> This model (DDPM) achieves FID that is better than any other GAN, and it is the likelihood-based model. Therefore, for coherence of the comparison, we excluded the WGAN and we decided to exclude any GAN-based model from the table. Nevertheless, we are thankful for providing a couple of relevant papers.
> Due to the double-blind policy, we cannot explain all details of how the project was carried out. Unfortunately, due to some circumstances, we were unable to calculate the scores.
>
> **Point No other models on CelebA**
> It is rather surprising, but it is very hard to find a paper that reports bpd on CelebA. There is a vast of papers that use this dataset, but they provide only samples or FID scores. Eventually, after an extensive literature search, we were able to find a score (bpd) in the RealNVP that we included in the new version of the paper. We fully agree that having at least one baseline positions our results better.
>
> **Point on Imagenette64**
>
> We have indicated in the paper that it is a version of the ImageNet dataset provided by FastAI. However, it is obtained in a slightly different manner than the ImageNet64. First, downscaling is done differently. Second, it is a subset of ImageNet64. We did our best, but we were unable to find any other paper that used Imagenette for density estimation.
> We have used Imagenette instead of ImageNet due to limited computational resources. We are aware that this is not the best excuse, but we decided that it is better to provide another dataset that consists of 64x64 images rather than working on simpler models like MNIST or Omniglot that contain 28x28 images.

---

### Official Review · AnonReviewer4 · 2020-10-26
**Difficult to follow**

**Rating:** 5
**Confidence:** 1

**Review:**


## Summary

The paper presents a self-supervised variational auto-encoder called selfVAE. The work proposes the use of downscaling and edge detection as simpler representations of the input images to be reconstructed. The model should then learn to improve the low dimensional approximations to recover the higher dimensional ones in a hierarchical fashion.

## Quality & Clarity

The paper is generally quite difficult to follow and the purpose, contributions and experiments are not presented clearly enough. The figures are not discussed in order, and the paper often references figures that are far away.

There are a number of grammatical errors in the paper.

## Outcome

The message of the paper generally was quite unclear and it could do with restructuring to assist readers.

---

> ### Author Response · Authors · 2020-11-16
> **Response to AnonReviewer4**
>
> The only suggestion following from this review is that the paper is hard to follow. We have rewritten many parts of the paper and we do hope that it is more readable now.

---

### Official Review · AnonReviewer2 · 2020-10-28
**Interesting work, but some details need to be further clarified.**

**Rating:** 6
**Confidence:** 4

**Review:**

This paper targets richer and higher-quality generation with VAE. Two techniques are adopted to achieve the goal: 1). bijective model to enrich data generation with flexible prior. 2). presenting compressed variants of the input data, i.e. self -supervision as additional condition $y$, for reconstruction. The two techniques interact through a hierarchical sampling process, $... y\sim p(y|u)\rightarrow z\sim p(z|u,y)$, thus benefits VAE generation with data-dependent prior and condition generation.

The idea novel and reasonable. the paper is clearly presented. Here are some of my concerns.

1.  The author specifically argues the transformation $x \rightarrow y$ to be 'non-trainable', i.e. the mapping between x and y is deterministic.  BUT, will modeling $q(x|z,y)$ with discretized logistic distribution, affect the generation quality, since the likelihood is classicly assumed to be Gaussian distributed?

2. The HIERARCHICAL SELF-SUPERVISED VAE is presented here to show the model can adopt multi-scaling information to benefit generation step-by-step. However, I  am afraid, in this way, the inference would be much difficult since the flow-based bijective operation is hard to train already.

3. Is the conditional information, e.g. the sketches, also need in the test phase? or unconditional generational setting is adopted here.

4.  It seems the experiments are not conducted on High-quality datasets. To me, the presented results can not obviously demonstrate the achievements of the model.

5. Can you please explain the connection between your self-supervised VAE to the general conditional VAE model in [1].

[1] Sohn, Kihyuk, Honglak Lee, and Xinchen Yan. "Learning structured output representation using deep conditional generative models." Advances in neural information processing systems. 2015.

---

> ### Author Response · Authors · 2020-11-16
> **Response to AnonReviewer2**
>
> First of all, we would like to thank you for your kind words and insightful comments. Please find our response below:
>
> **Point 1**
>
> This is true that in many papers the decoder, i.e., $p(x|\ldots)$, is modeled as the Gaussian distribution. Unfortunately, this is not correct in the case of images that are represented by integers in $\{0, 1, \ldots, 255\}$. Variational Auto-Encoders are so-called prescribed latent variable models in which all probability distributions must be defined upfront. Moreover, all distributions must be properly chosen for random variables. By “properly” we mean appropriately to the values that a random variable can take. For instance, Gaussian distribution, which has the support between $-\infty$ and $+ \infty$, mustn’t be used for integers. We are aware that it is a common practice to use Gaussians for images, which results in the MSE loss in deep learning packages, but it is not theoretically grounded. Therefore, we used the discretized logistic distribution, which is appropriate for integers, following other papers in the literature, e.g.:
> - Salimans, T., Karpathy, A., Chen, X., & Kingma, D. P. (2017). Pixelcnn++: Improving the pixelcnn with discretized logistic mixture likelihood and other modifications. ICLR 2017.
>
> **Point 2**
>
> It is a very interesting remark. In the very beginning of the project, we checked how selfVAE works while being trained in two stages (i.e., for one minibatch, first the encoder and the decoder were updated, and then the bijective prior was updated) against selfVAE being trained in a standard manner (i.e., all weights were updated at once). It turned out that there was no difference, and even the standard training seemed to give better results. Therefore, we used the standard training.
> The flow-based model is considered hard to train when the target data are of high-dimensionality (i.e. natural images). However, in our case, the target distribution is of lower dimensionality (1024 dimensions) which allows up to incorporate the best of both worlds; a smaller, fast data-dependent prior that does not collapse (please see for ablation study). Given the deterministic transformations, we experienced that the model scales easily while incorporating the bijective prior, without any effort to tune any hyper-parameters. To conclude, given this very reasonable remark, our model was specially designed to scale efficiently and unhesitatingly.
> Moreover, another possible explanation of this result is the following. A vanilla hierarchical VAE can suffer from posterior collapse because z’s on top obtain data that are already processed through multiple stochastic layers. However, in our approach, we provide processed data through deterministic and discrete transformations (e.g., downscaled images), therefore, on top, we have a direct connection to data, and learning a bijective prior p(u) is less problematic.
>
> **Point 3**
>
> In our opinion, the strength of our approach is that we can use either a conditional generation or an unconditional generation. Therefore, regarding your question, we can use both. In the paper, we highlight whether we present unconditional or conditional samples.
>
> **Point 4**
>
> Partially, we agree with this comment. Applying our approach to CIFAR10 has a rather limited effect because the images are very small (32x32). However, results on 64x64 images (CelebA, Imagenette) are promising and indicate that both multiple downscaling and sketching have a positive effect. Nevertheless, we would love to apply our approach to larger images, but we simply did not possess enough computational power to accomplish that. We are almost certain that our approach would shine even more on 256x256 or larger images.
>
> **Point 5**
>
> Thank you for pointing out the paper. We are aware of this paper since it is one of the first successful papers on modeling both image and its label using VAEs. The main difference between the conditional VAE (CVAE) and our selfVAE lies in the quantities we model. In our case, we focus on y that is a transformation of $x$, $y=d(x)$, while in the CVAE case, y is a label (or a segmentation). Further, CVAE models the conditional likelihood, $p(y|x)$, while we model the joint distribution, $p(x,y)$. We have also rewritten our model slightly to make this distinction even more apparent.

---

### Decision · Program_Chairs · 2021-01-07
**Final Decision**

**Decision:**

Reject

**Comment:**

Reviewers appreciated the model and the ideas presented and found them very interesting.

The main reason for rejection is the extent of the empirical work.  Unfortunately, and I think what is a bad sign for the ICLR community, the authors could not do adequate empirical work due to their computational resources.  Not belonging to an organisation with extensive computational resources myself, I am in strong symparthy with the authors, though I do not see any way this can be satisfactorily accounted for in reviewing.  Several reviewers commented on the datasets, the extent of evaluations, and the comparisons made with prior work.  For instance, the small CIFAR10 images are not ideal to demonstrate the technique and comparative results with the other data sets are limited.

The reviewers had a number of concerns on the theoretical work and these were well discussed by the authors.

In summary, this is promising research but needs more empirical work.